# Integrating Multiple Omics Identifies *Phaeoacremonium rubrigenum* Acting as *Aquilaria sinensis* Marker Fungus to Promote Agarwood Sesquiterpene Accumulation by Inducing Plant Host Phosphorylation

Juan Liu,[a] Tianxiao Li,[a] Tong Chen,[a] Jiaqi Gao,[a] Xiang Zhang,[a] Chao Jiang,[a] Jian Yang,[a] Junhui Zhou,[a] Tielin Wang,[a] Xiulian Chi,[a] Meng Cheng,[a] Luqi Huang[a]

[a]National Resource Center for Chinese Materia Medica, China Academy of Chinese Medical Sciences, Beijing, China

**ABSTRACT** The present study aimed to explore the factors that promote persistent agarwood accumulation. To this end, we first investigated the morphological changes and volatile compound distribution in five layers of "Guan Xiang" agarwood. The agarwood-normal transition layer (TL), an essential layer of persistent agarwood accumulation, showed clear metabolic differences by microscopy and GC-MS analysis. Microbiome analysis revealed that *Phaeocremonium rubrigenum* was the predominant biomarker fungus in the TL of "Guan Xiang" agarwood samples. Among the seven isolated fungi, *P. rubrigenum* exhibited a significantly heightened ability to induce the production in *Aquilaria sinensis* seedlings, especially for sesquiterpene. Tracing the proteome profile changes in *P. rubrigenum*-induced *A. sinensis* calli for 18 ds showed that the fungus-induced sesquiterpene biosynthesis increased mainly through the mevalonate (MVA) pathway. Specifically, the phosphorylation modification level, instead of the protein abundance of transcription factors (TFs), showed corresponding changes during sesquiterpene biosynthesis, thus indicating that induced phosphorylation is the key reason for enhanced sesquiterpene production.

**IMPORTANCE** Agarwood is an expensive resinous portion derived from *Aquilaria* plants and has been widely used as medicine, incense, and perfume. The factors involved in steady agarwood accumulation remain elusive. Our current study suggests that as a TL marker fungus, *P. rubrigenum* could persistently promote agarwood sesquiterpene accumulation by inducing phosphorylation of the TFs-MVA network in *A. sinensis*. Moreover, our work provides strategies to improve agarwood industry management and sheds light on the potential molecular mechanisms of plant adaptation to native microbial conditions.

**KEYWORDS** agarwood, microbiome, *Phaeocremonium rubrigenum*, sesquiterpenes, protein phosphorylation, transcription factors, multi-omics

Living organisms are part of a highly interconnected web of interactions. Vast networks of microbes play important roles in stimulating host plant growth, antagonizing pathogens, tolerating stress, and controlling plant diseases (1, 2). This could be exploited to achieve high crop yields and agriculturally relevant medicinal plants. However, the association between the microbiome and the formation of herbal resin, which is regarded as the result of plant adaptation to native microbial conditions, is rarely investigated.

Agarwood is a resinous and fragrant wood that is an essential ingredient of some exclusive perfumes and is extensively used as medicine and incense across Asia, the Middle East, and Europe (3, 4). The annual global agarwood market is valued at 6 to 8 billion USD, not including the proportion of unrecorded trade (5, 6). Agarwood is naturally produced as a response to microbial or insect attacks on tropical trees of the genus *Aquilaria* (7). Additionally, natural agarwood formation can take up to 10 years. Due to the slow and infrequent formation of

Address correspondence to Juan Liu, juanliu126@126.com, Tong Chen, chent@nrc.ac.cn, or Luqi Huang, huangluqi01@126.com.

The authors declare no conflict of interest.

agarwood and the overexploitation of wild resources, *Aquilaria* spp. populations are declining dramatically. All species of the *Aquilaria* genus are listed in Appendix II of the Convention on International Trade in Endangered Species of Wild Fauna and Flora (CITES, http://www.cites .org). This includes *Aquilaria sinensis* (Lour.) Gilg, the only certified source of agarwood listed in the 2020 Chinese Pharmacopoeia (8). The mechanism of agarwood formation has been investigated because of its rarity and importance. Previous studies have found that the main mechanism underlying agarwood formation involves plant defense mechanisms that initiate the production of resin-rich sesquiterpenes and 2-(2-phenylethyl) chromone derivatives (9–12). However, the mechanisms promoting persistent agarwood accumulation, which could effectively thicken the agarwood layer and further improve the high yields of agarwood, remain largely unknown.

With an increasing number of *Aquilaria* plantations, various technologies have been developed to ensure agarwood yield stability (6). Agarwood-inducing technologies employing natural factors are more likely to produce high-quality agarwood (13, 14). "Guan Xiang," the agarwood produced in Dongguan, Southern China, has a long history of harboring *A. sinensis*, as early as the reign of Tang and Song Dynasties, and has inherited the natural ability to produce a thick layer of agarwood (15), usually of ~0.5-2 cm thickness. Therefore, we use "Guan Xiang" as our research material to investigate the potential approach to promote persistent agarwood accumulation.

Fungal infections in *Aquilaria* host trees have been recognized as the main cause of natural agarwood formation in the last century (16–18). Fungal strains used for agarwood induction are obtained from natural source and are often safe to handle and environmentally friendly (6). Only 8% of the fungi isolated from *Aquilaria* trees have been explored for their active role in agarwood enhancement (7). These include *Lasiodiplodia theobromae*, *Trichoderma* sp., *Cladosporium* sp., *Penicillium* sp. and *Fusarium* sp. (19–22), most of which were isolated from healthy *Aquilaria* tissue or whole agarwood samples. However, there is still no detailed profile of microbiome distribution in separated layers of argarwood, especially in "Guan Xiang," which could help us screen the key fungal strain promoting persistent agarwood accumulation.

In this study, we stratified the "Guan Xiang" agarwood into five layers based on agarwood abundance and profiled the phenotypic, metabolic, and fungal diversity of layers. Our results showed that fungal diversity is negatively correlated with the abundance of sesquiterpenes and chromones. Several marker species were enriched in the agarwood-normal transition layer. Of these *Phaeoacremonium rubrigenum* showed the highest enrichment. Treatment with the isolated *P. rubrigenum* persistently promoted agarwood sesquiterpene and chromone accumulation, which are the two major componenets of agarwood. Compared to the six other isolated fungi enriched in other layers, *P. rubrigenum* exhibited a significantly heightened ability to induce sesquiterpene production in *A. sinensis*. Proteome analysis showed that the MVA pathway is mainly induced by sesquiterpene accumulation, and high phosphorylation-activated transcription factors (TFs) form an upstream response network. This study integrates microbiome, proteome, and phosphoproteomics to profile the fungal composition in five layers of agarwood and identified that *P. rubrigenum* could harmlessly and continuously promote sesquiterpene accumulation by inducing the phosphoralated TF-MVA network in *A. sinensis*.

## RESULTS

**Morphological changes and volatile oil distribution in different layers of agarwood formed by physically wounding *A. sinensis*.** Agarwood samples were collected from a 50-year-old *A. sinensis* tree, also known as "Guanxiang" in Dongguan 5 years after employing the physical wounding method (7). Agarwood samples were divided into five regions at different depths from the wound surface by visual inspection: a decay layer (DL), a decay-agarwood transition layer (DA), an agarwood layer (AL), an agarwood-normal transition layer (TL), and a normal layer (NL) (Fig. 1A). Starch and polysaccharides covering ray parenchyma cells were observed from the NL to the TL, but no starch and polysaccharides were found in the other layers (Fig. S1A). In contrast, numerous brown resin-like materials appeared in the AL and TL, but none were observed in the NL (Fig. S1A). Under DAPI staining, only the nuclei

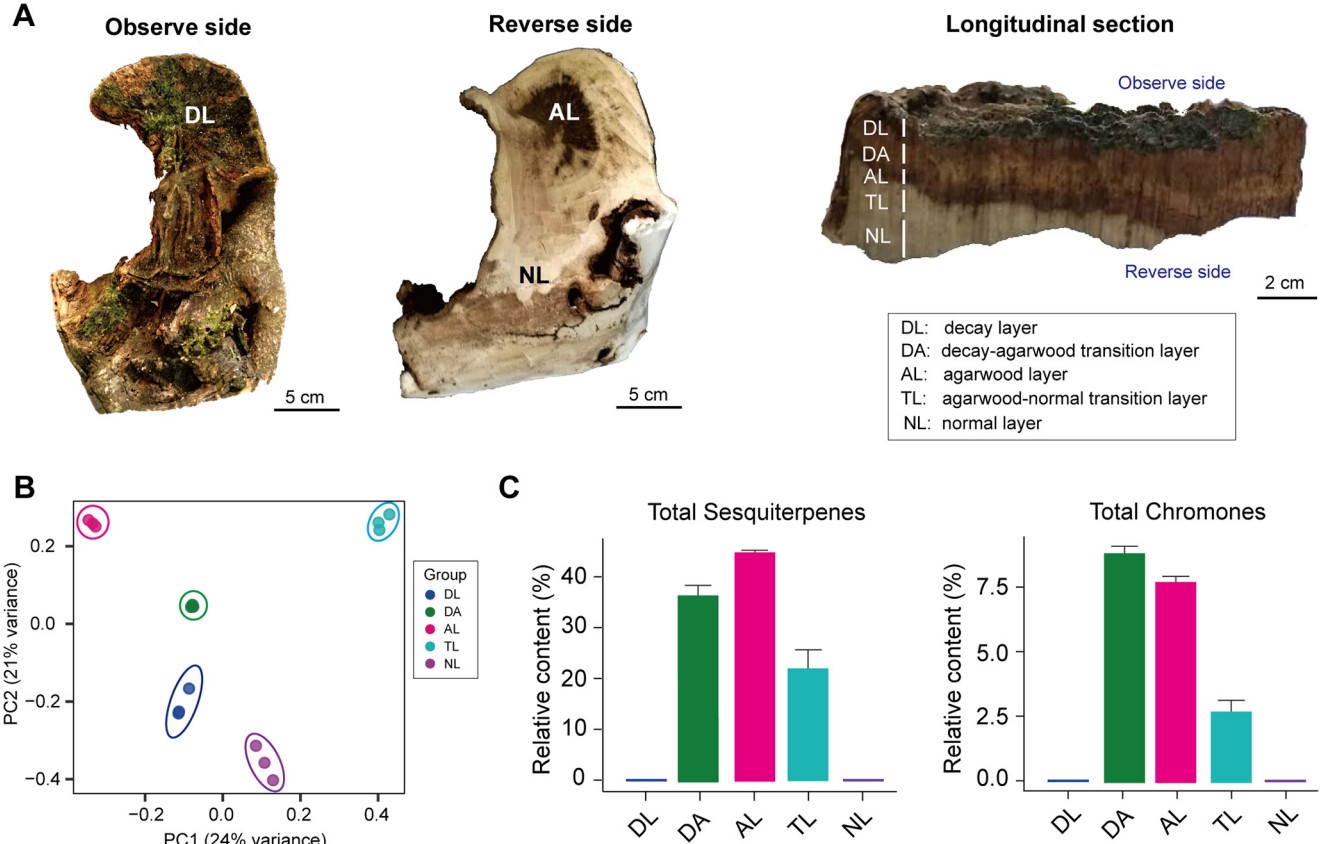

**FIG 1** Different composition of volatiles in five agarwood layers. (A) Samples from different layers of agarwood (DL, decay layer; DA, decay-agarwood transition layer; AL, agarwood layer; TL, agarwood-normal transition layer; NL, normal layer). Observe side, the outer side of the agarwood samples showing the DL. Reverse side, the inner side of the agarwood samples showing the NL. (B) Unconstrained PCA (for principal components PC1 and PC2) depicting the volatile metabolite differences among different layers of agarwood. (C) Relative content of total sesquiterpenes and total chromones in different layers of agarwood by GC-MS analysis. *n* = 3.

of the living cells became fluorescent. The shape of the nuclei varied according to the location of the cells. Living cells in the NL contained round-shaped nuclei, whereas elliptical nuclei were observed in living cells from the TL (Fig. S1A). Nuclei were not observed in the AL and DA cells, suggesting that most AL and DA cells were dead (Fig. S1A). The above structural characteristics of the agarwood samples in this study were consistent with those previously reported (23). We characterized the distribution patterns of starch, polysaccharides, brown resin-like materials, and live cells across the different layers of wounded *A. sinensis* wood (Fig. S1B). All metabolites in each layer were profiled, and the principal-component analysis (PCA) results showed remarkable differences in the composition of metabolites in each layer (Fig. 1B). Agarwood contains mainly sesquiterpenes and 2-(2-phenethyl) chromones. We analyzed the distribution of sesquiterpenes and phenylethyl chromone derivatives in volatile oil across the different layers of wounded trees using GC-MS (Table S1). Sesquiterpenes were mainly detected in TL, AL, and DA; AL and DA had the highest concentrations of sesquiterpenes and chromones across the five layers of wounded *A. sinensis* trees, followed by TL (Fig. 1C and Table S1). This was consistent with the distribution of the brown resin-like materials. A previous study showed that agarwood resin formed and accumulated in the living parenchyma cells of the interxylary phloem and xylem rays (23). The TL is a key region connecting the AL and NL with the living parenchyma cells, which have the potential to accumulate resin continuously (Fig. S1). In addition, both our results and a previous report revealed that the metabolism in the TL changed significantly, such as starch, polysaccharide, sesquiterpenes, and chromones (Fig. S1 and Table S1) (23), which suggest that the TL is a key region for further investigated.

**Composition of fungal communities in different layers of agarwood.** To investigate the role of fungal partners in continuous agarwood accumulation, fungal communities associated with different layers of agarwood were investigated by high-throughput sequencing. Approximately 80,055-80,335 high-quality reads were obtained from agarwood samples, which belonged to 6 phyla, 27 classes, 74 orders, 135 families, 189 genera, and 247 species of fungi, with 305 to 660 OTUs detected at 97% similarity (Fig. S2A and Table S2). The alpha diversity of fungal species observed was significantly higher in DL than in the other four layers ($P < 0.05$) (Fig. 2A and Table S3). The community richness indices, including ACE, Chao 1, and PD whole tree, indicated that the three layers containing agarwood resin (DA, AL, TL) possessed the lowest richness. In contrast, DL possessed the richest fungal community (Fig. 2A and Table S3). The Shannon index indicated the lowest fungal diversity in AL and TL, suggesting that the fungal specificities of AL and TL were strong (Fig. 2A and Table S3). In addition, differences in the fungal communities between the samples were clustered using nonmetric multidimensional scaling (NMDS) and principal coordinate analysis (PCoA). This also revealed that the fungal communities in the DL and NL were different from those in the other layers that contained resin-like materials (Fig. 2B and Fig. S2B). Of the six detected fungal phyla, Ascomycota was the most dominant in all layers of agarwood (Fig. S3A). Among the 10 most abundant fungal classes, orders, and families, Dothideomycetes, Pleosporales and Incertae sedis Pleosporales were dominant in the AL (Fig. S3B, C, D), but Sordariomycetes, Diaporthales, and Togniniaceae were dominant in the TL (Fig. S3B, C, D). The dominant genus of AL and TL out of the 10 most abundant fungal genera was *Phaeoacremonium* (Fig. S3E), and *P. rubrigenum* was the most abundant fungal species in the AL and TL (Fig. S3F). Linear discriminant analysis (LDA) effect size (LEfSe) also revealed that *P. rubrigenum* was the TL biomarker with a statistical difference (Fig. 2C). The correlation network showed that the Ascomycota phylum had the strongest correlation with sesquiterpenes, especially the *Phaeoacremonium* order and *P. rubrigenum* (Fig. S4), suggesting that *P. rubrigenum* may play a key role in persistent agarwood accumulation.

***P. rubrigenum* promotes the persistent accumulation of volatiles in agarwood.** To further characterize the role of *P. rubrigenum* in persistent agarwood accumulation (Fig. 3A), we isolated this fungus in the TL, which can be obtained in a relatively environment-friendly manner from natural sources. However, *P. rubrigenum* cannot be isolated by incubation on potato dextrose agar (PDA). Therefore, we isolated *P. rubrigenum* onto malt extract agar (MEA) and confirmed the identity by phenotypic and molecular characterization (Fig. 3B and C; Fig. S5). No sesquiterpenes and farnesyl pyrophosphate (FPP, the precursor of sesquiterpenes) were detected in either the hyphae or fermentation media of *P. rubrigenum* by GC-MS analysis (Fig. S6), suggesting that the interaction between *A. sinensis* and *P. rubrigenum* could play a key role in promoting the persistent accumulation of volatiles in agarwood. To investigate their interaction, 1-year-old *A. sinensis* seedlings were holed and infected with or without *P. rubrigenum*. *A. sinensis* stems were collected every 5 days over a 30-day treatment period. Compared with the control group (samples holed without *P. rubrigenum*), the sesquiterpenes and chromone were only detected in the infected group using GC-MS (Table S4). The relative contents of total sesquiterpenes and chromones after 30 days of treatment were 5.572% and 0.260%, respectively (Fig. 3D). Further analysis showed that the total contents of sesquiterpenes and chromones increased considerably after 15 days of *P. rubrigenum* infection (Fig. 3E). To verify the activity of *P. rubrigenum*, six fungi, mainly distributed in other layers of agarwood, were isolated and used to treat *A. sinensis* seedlings. *Lasiodiplodia theobromae*, reported in a previous study (20), was mainly distributed in DL. *Trichoderma harzianum* and *Trichoderma atroviride*, belonging to the *Trichoderma* genus reported in a previous study (21), were mainly distributed in DL and NL. *Cladosporium cladosporioides*, *Cladosporium parahalotolerans*, and *Penicillium janthinellum*, which have been reported in previous studies (19, 22), were mainly distributed in the NL. Among the six fungi isolated from agarwood, *T. harzianum*, *T. atroviride*, and *C. cladosporioides*, and most importantly *L. theobromae*, induced a greater production of total chromones than *P. rubrigenum*. Compared to the above six fungi, *P. rubrigenum*

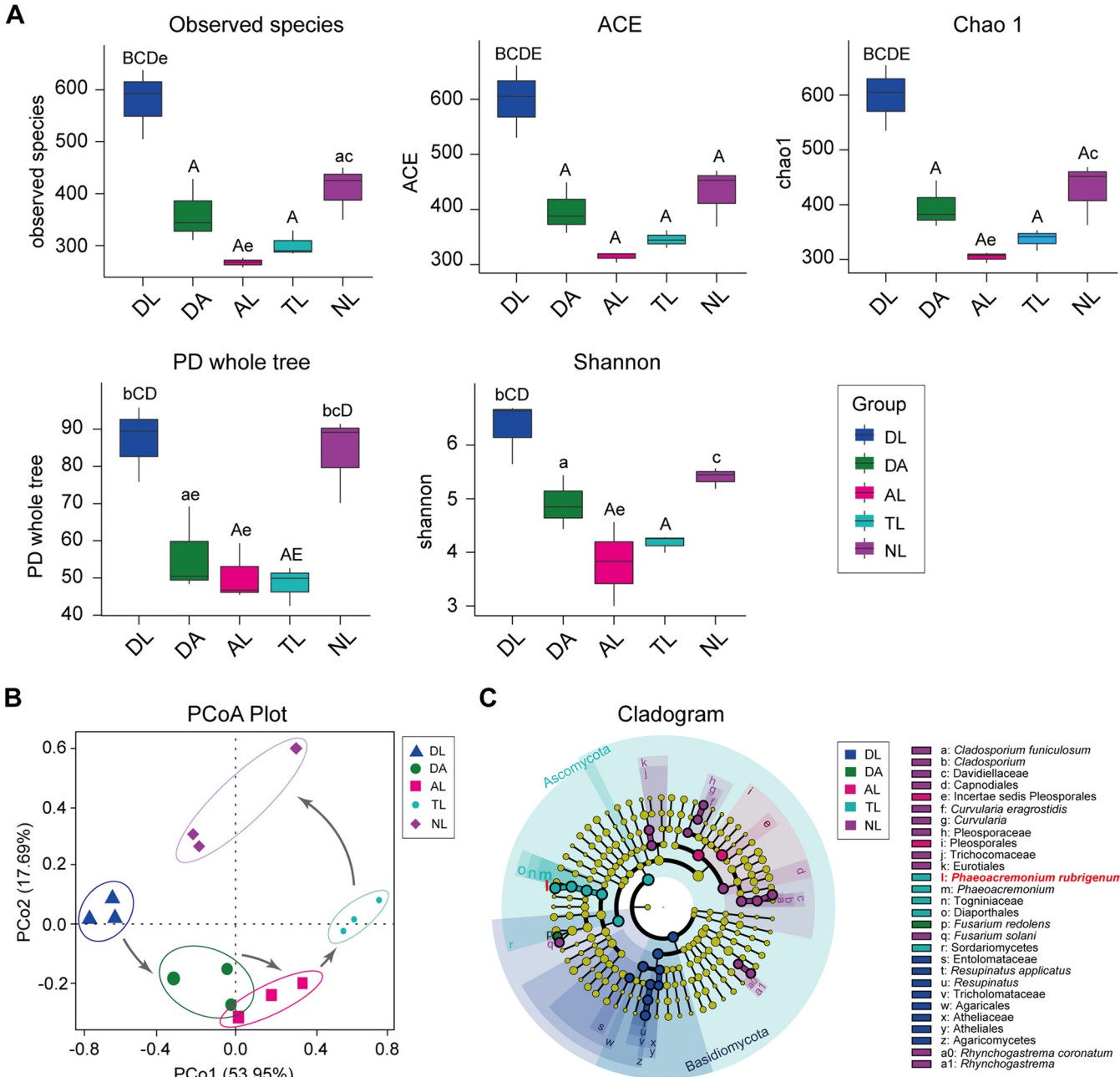

**FIG 2** Different composition of fungal communities in five agarwood layers. (A) Boxplot showing alpha diversity estimate, including Observed species, ACE, Chao1, PD whole tree, and Shannon of fungal communities of each layer of agarwood. Cross-sample differences were analyzed using Tukey analysis (Table S3). Significant differences (*P*-value < 0.05) across different compartments were indicated with lowercase letters (a, significant difference between DL and the other layers using Tukey analysis, *P* value < 0.05; b, significant difference between DA and the other layers using Tukey analysis, *P* value < 0.05; c. significant difference between AL and the other layers using Tukey analysis, *P* value < 0.05; d. significant difference between TL and the other layers using Tukey analysis, *P* value < 0.05; e. significant difference between NL and the other layers using Tukey analysis, *P* value < 0.05). Significant differences (*P* value < 0.01) across different compartments were indicated with uppercase letters (A, significant difference between DL and the other layers using Tukey analysis, *P* value < 0.01; B, significant difference between DA and the other layers using Tukey analysis, *P* value < 0.01; C. significant difference between AL and the other layers using Tukey analysis, *P* value < 0.01; D. significant difference between TL and the other layers using Tukey analysis, *P* value < 0.01; E. significant difference between NL and the other layers using Tukey analysis, *P* value < 0.01). DL, decay layer; DA, decay-agarwood transition layer; AL, agarwood layer; TL, agarwood-normal transition layer; NL, normal layer. (B) Unconstrained PCoA (for principal coordinates PCo1 and PCo2) with weighted UniFrac distance depicting the fungal species composition structure differences among different layers of agarwood. (C) The LEfSe generated the taxonomic cladogram showing marker fungal species of each layer of agarwood. Each dot represents one taxon, with sizes indicating average abundances. The color of dots represents taxon marker belongings of groups. *Phaeoacremonium rubrigenum* was labeled with red color, indicating it was the biomarker of the TL.

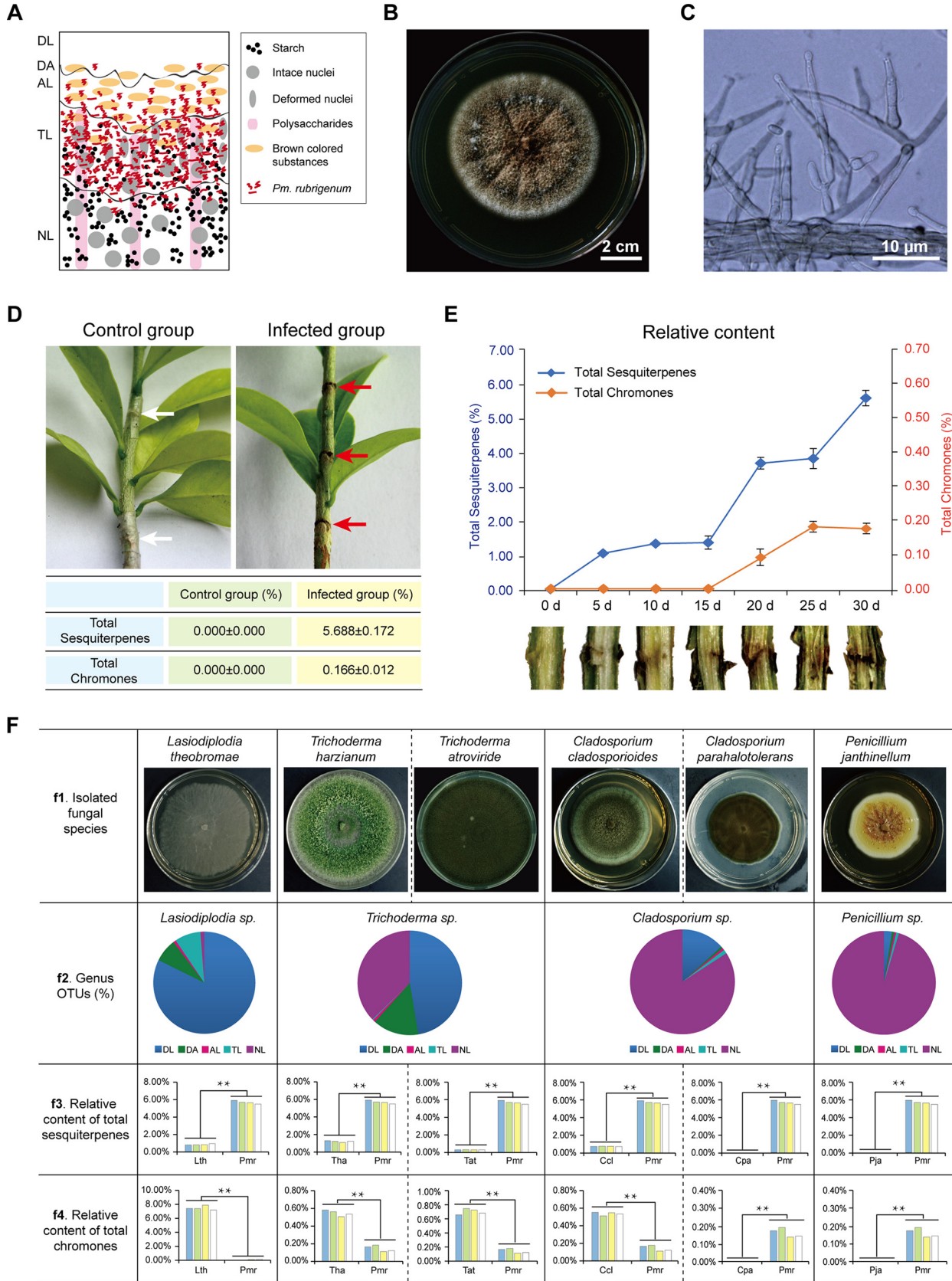

**FIG 3** Characterizing the role of *P. rubrigenum* in agarwood persistent accumulation. (A) Diagram of *P. rubrigenum* distribution patterns of storage starch, polysaccharides, brown resin-like materials, and nuclei in the different layers of agarwood. (B) Culture characteristics of *P. rubrigenum* on malt

had a heightened ability to induce the production of sesquiterpenes in *A. sinensis* seedlings (Fig. 3F). Our data demonstrated that *P. rubrigenum* promoted the persistent accumulation of volatile compounds in *A. sinensis* seedlings, especially sesquiterpenes, compared to the other six fungi.

**_P. rubrigenum_ induces the sesquiterpene biosynthesis of _A. sinensis_ by the proteome analysis.** To investigate the role of *P. rubrigenum* in the persistent accumulation of sesquiterpenes in *A. sinensis*, the volatiles of *A. sinensis* calli stressed by *P. rubrigenum* on different days were analyzed (Fig. 4A and B). Four sesquiterpenes were detected in *A. sinensis* calli: $\delta$-guaiene, $\alpha$-copaene, $\alpha$-guaiene, and nootkatene (Table S5). After 3 days of treatment, four sesquiterpenes were induced in *A. sinensis* calli (Fig. 4B). The 18 days treated *A. sinensis* calli displayed a 10 to 40 stronger level of sesquiterpenes than the 0-day calli (Fig. 4B). We then compared the protein abundance profiles of the treated *A. sinensis* calli using tandem mass tags (TMT) labeling technology based on nanoscale liquid chromatography-mass spectrometry analysis (LC-MS/MS). Principal-component analysis (PCA) indicated that the protein profiles in the 0-day control group were distinct from those in the *P. rubrigenum* treated groups, especially in the 18-day group (Fig. 4C). Differentially expressed proteins (DEPs) were clustered into six groups according to protein abundance profiles, and 58.9% of DEPs were in clusters IV and V, which were more abundant in the *P. rubrigenum* treated groups than in the controls (Fig. 4D). Additionally, the differential abundance of enzymes involved in the sesquiterpene biosynthesis pathway among the groups was analyzed. After normalization, the results of the relative quantitative protein abundance showed that the enzymes in the mevalonate (MVA) pathway were significantly upregulated. In contrast, the enzymes in the methylerythritol 4-phosphate (MEP) pathway were significantly downregulated (Fig. 4E and Table S6). Downstream enzymes, including farnesyl diphosphate synthase (FPS) and guaiene synthases (SesTPS1 and SesTPS2), were also significantly upregulated (Fig. 4E and Table S6). Our results suggest that *P. rubrigenum* mainly induces sesquiterpene biosynthesis in *A. sinensis* via the MVA pathway.

**_P. rubrigenum_ could regulate the phosphorylation modification levels of transcription factors in _A. sinensis_.** A growing number of findings have demonstrated that transcription factors (TFs) and plant immunity signaling proteins are involved in the regulation of sesquiterpene biosynthesis of *A. sinensis* (6, 12, 24, 25). Therefore, we analyzed the TF protein levels. A total of 12 common TF families (i.e., bHLH, ERF, WRKY, MYB, bZIP, NAC, MADS, DOF, HD, TCP, SBP, and GRAS) were identified. These TFs play essential roles in plant stress resistance, secondary metabolism, and plant development (Table S7). However, the protein levels of 71% of the TFs did not change significantly, and only three TFs (ARR2, ALR2, and NAC17) were upregulated after 3 or 18 days of *P. rubrigenum* induction (Table S7). Therefore, there may be another TF regulatory mechanism involved in sesquiterpene biosynthesis promoted by *P. rubrigenum*.

Protein phosphorylation is one of the most common posttranslational modifications in plants, and the phosphorylation status of a protein can have profound effects on its activity and interactions with other proteins (26, 27). Therefore, we analyzed protein phosphorylation levels in the calli of *A. sinensis* before and after *P. rubrigenum* stress. The phosphorylated proteins were annotated in the molecular function of protein and nucleic acid binding, containing TFs (Fig. 5A). InterPro (IPR) annotation of phosphoproteomics revealed that the protein

**FIG 3** Legend (Continued)

extract agar (MEA) medium. (C) Microscopic observations of *P. rubrigenum* mycelia from colonies cultivated on Malt Extract Agar (MEA) medium. (D) The total sesquiterpene and chromone content in 1-year-old *A. sinensis* seedlings after treatment with the holing technique with or without *P. rubrigenum* for 30 days. n = 4. (E) Time-series accumulation of relative abundance of sesquiterpene and chromone content after *P. rubrigenum* treatment within 30 days. Each value represents the mean of four replicates ± standard deviation (SD). n = 4. (F) Comparing relative content of total sesquiterpene and chromone accumulation differences in 1-year-old *A. sinensis* seedlings after treatment with six other fungal species for 30 days, including *L. theobromae* (Lth), *T. harzianum* (Tha), *T. atroviride* (Tat), *C. cladosporioides* (Ccl), *C. parahalotolerans* (Cpa), and *Penicillium janthinellum* (Pja). f1: Culture characteristics of the above six fungal species isolated from agarwood on potato dextrose agar (PDA) medium. f2: Pie charts showing the relative abundance of the above fungal genera in each agarwood layer, including *Lasiodiplodia* sp., *Trichoderma* sp., *Cladosporium* sp., *Penicillium* sp. f3: Comparing the total sesquiterpene content in 1-year-old *A. sinensis* seedlings after treatment using the holing technique with *P. rubrigenum* (Pmr) and other isolated fungi (Lth, Tha, Tat, Ccl, Cpa, Pja) for 30 days. n = 4. **, *P* value < 0.01. f4: Comparing the total chromone content in 1-year-old *A. sinensis* seedlings after treatment using the holing technique with *P. rubrigenum* (Pmr) and other isolated fungus (Lth, Tha, Tat, Ccl, Cpa, Pja) for 30 days. n = 4. **, *P* value < 0.01.

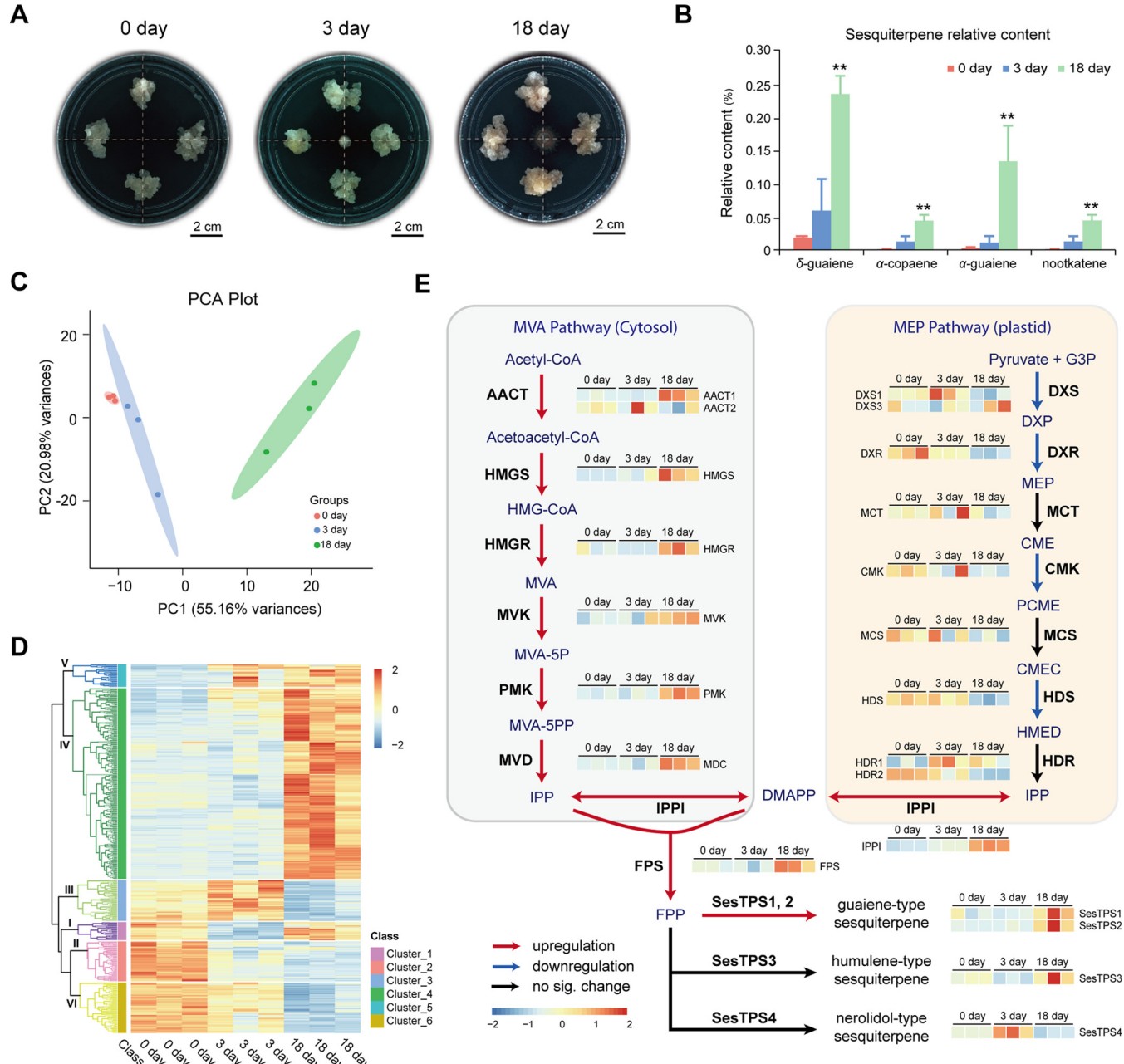

**FIG 4** Proteome analysis showing protein abundance profile changes along with *P. rubrigenum* induced sesquiterpene biosynthesis of *A. sinensis*. (A) Status of *A. sinensis* calli during *P. rubrigenum* inducing from 0 to 3 and 18 days. (B) Sesquiterpene relative content in the above three groups, including 0 day, 3 day and 18 day *P. rubrigenum* stressed *A. sinensis* calli, detected by GC-MS. *n* = 3. (C) Principal-component analysis (PCA) showing protein abundance profile differences among three induced *A. sinensis* calli groups. (D) Clustering of differentially expressed proteins (DEPs) in three induced *A. sinensis* calli groups. (E) The relative abundance of enzymes involved in sesquiterpene biosynthesis by using quantitative proteomics analysis, including mevalonate (MVA) pathway and methylerythritol 4-phosphate (MEP) pathway. AACT, acetyl-CoA C acetyltransferase; CME, 4-(cytidine 5′-diphospho)-2-C-methyl-D-erythritol; CMEC, 2-C-methyl-D-erythritol 2,4-cyclodiphosphate; CMK, 4-(cytidine 5′-diphospho)-2-C-methyl-D-erythritol kinase; DXP, 1-deoxy-D-xylulose 5-phosphate; DXR, 1-deoxy-D-xylulose 5-phosphate reductoisomerase; DXS, 1-deoxy-D-xylulose 5-phosphate synthase; FPS, farnesyl diphosphate synthase; G3P, glyceraldehyde 3-phosphate; HDS, 4-hydroxy-3-methylbut-2-enyl diphosphate synthase; HDR, 4-hydroxy-3-methylbut-2-enyl diphosphatereductase; HMED, 4-hydroxy-3-methylbut 2-enyl diphosphate; HMG-CoA, hydroxyl methyl glutaryl-CoA; HMGS, hydroxyl methylglutaryl-CoA synthase; HMGR, hydroxyl methyl glutaryl-CoA reductase; IPP, isopentenyl diphosphate; IPPI, isopentenyl diphosphate isomerase; MCS, 2-C-methyl-D-erythritol 2,4-cyclodiphosphate synthase; MCT, 2-C-methyl-D-erythritol-4-phospate cytidylyltransferase; MEP, 2-C-methyl-D-erythritol 4-phosphate; MVA, mevalonate; MVA-5P, mevalonate-5-phosphate; MVA-5PP, mevalonate-5-diphosphate; MVD, mevalonate pyrophosphate decarboxylase; MVK, mevalonate kinase; PCME, 2-phospho-4-(cytidine 5′-diphospho)-2-C-methyl-D-erythritol; PMK, 5-phosphomevalonate kinase; SesTPS1 and SesTPS2, guaiene synthases; SesTPS3, humulene synthase; SesTPS4, nerolidol synthase. *n* = 3.

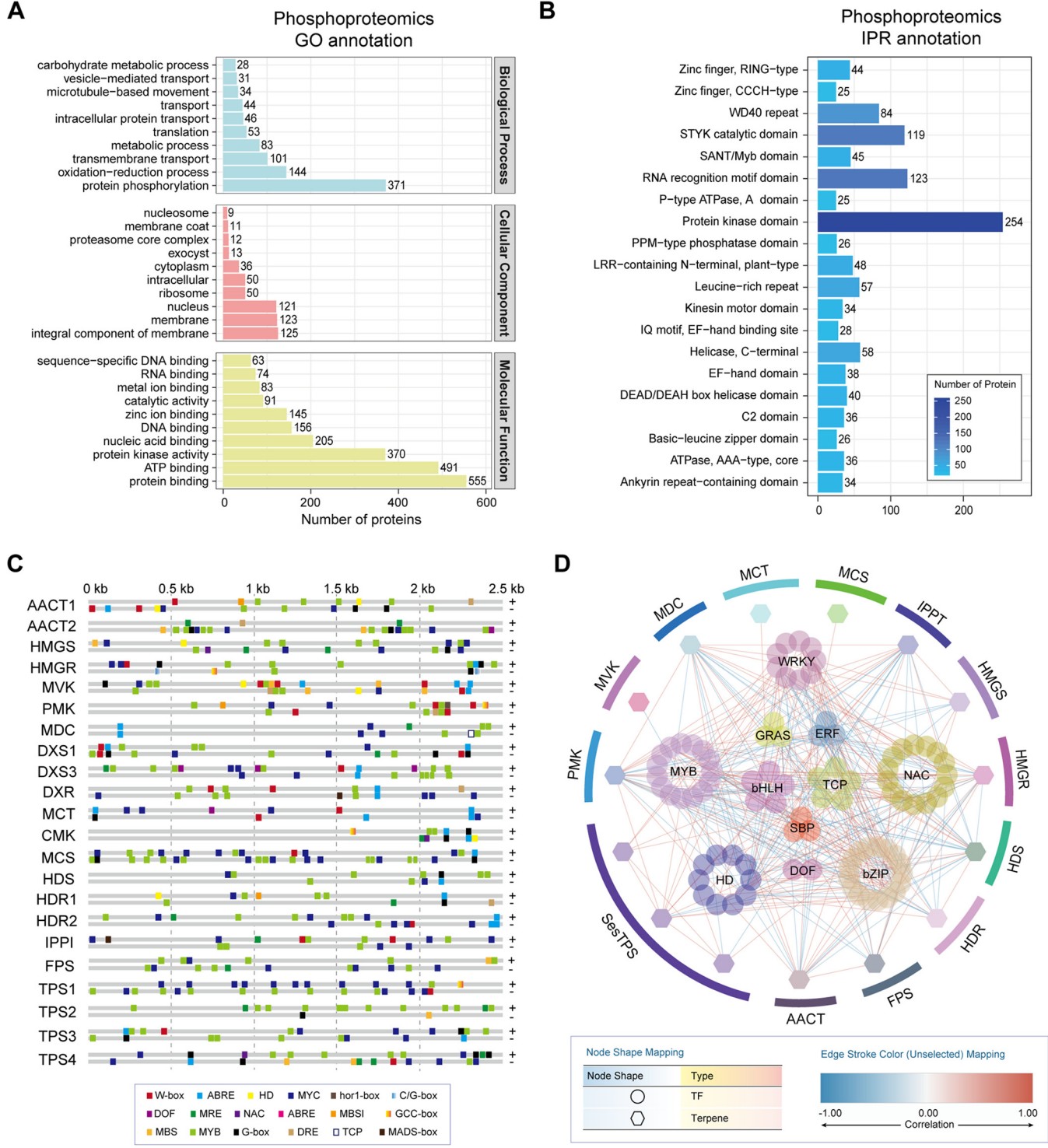

**FIG 5** Phosphoproteomics analysis of the phosphorylation modification levels in different *A. sinensis* calli groups. (A) Gene ontology (GO) annotation of phosphoproteomics in *A. sinensis* calli. (B) InterPro (IPR) annotation of phosphoproteomics in *A. sinensis* calli. (C) *Cis*-acting elements bound by TFs in the promoter sequences of the genes involved in agarwood sesquiterpene biosynthesis. +, forward sequence; -, reverse sequence. (D) Correlation analysis of transcription factors with pathway enzymes. Pathway enzymes are represented by hexagons and transcription factors by circles. The line indicates the nature of the correlation: red for a positive correlation and blue for a negative correlation.

kinase domain, including RLKs, MAPK cascades, and CPKs involved in the plant immunity signaling pathway, was the largest domain of phosphorylated proteins (Fig. 5B).

The protein phosphorylation levels of 52.9% TFs showed significant changes, including 20% phosphorylation-upregulated TFs and 32.9% phosphorylation-downregulated TFs,

especially in the MYB, WRKY, and bZIP families (Fig. S7 and Table S8). To further investigate the potential relationship between phosphorylated TFs and sesquiterpene biosynthetic genes, the promoter sequences of sesquiterpene biosynthetic genes were retrieved, and their regulatory sites of these promoters were identified by RegSite (https://www.softberry.com/) and PlantCARE (28). The promoters of sesquiterpene biosynthetic genes contained prolific types of *cis*-acting elements that could be bound by TFs, such as G-box, W-box, and GCC-box (Fig. 5C and Table S9). Among the above promoters, the MYB and MYC binding sites were the highest (Fig. 5C and Table S9). Correlation analysis was used to map the regulation network between the sesquiterpene biosynthetic pathway enzymes and phosphoralated TFs (Fig. 5D). The phosphoralated TF families that were highly correlated with pathway enzymes included phosphoralated MYB, bZIP, and WRKY TFs, which play a crucial role in plant growth and development, stress resistance, and secondary metabolism (Fig. 5D).

## DISCUSSION

The interactions between plants and fungi play key roles in regulating the host immune system and plant productivity, and knowledge regarding this relationship can help improve crop production. Agarwood is in high demand worldwide as a raw material for incense, perfume, and medicinal purposes. Immune responses of *Aquilaria* host trees and fungal infection generally seem to play essential roles in agarwood formation (6). This makes agarwood formation an excellent characteristic to shed light on the interaction between fungi and their plant host.

Previous studies have shown that *Phaeoacremonium* and *Fusarium* are the dominant genera representing > 50% of isolates from stem tissues of *A. sinensis* collected from rainforests in the Yunnan and Hainan provinces of China (29). Interestingly, *Phaeoacremonium* was also a member of the agarwood community in *A. malaccensis* (30). *Fusarium* can promote agarwood formation (31, 32). However, there are no reports on the roles of *Phaeoacremonium*, another dominant fungal genus, in agarwood formation and accumulation.

Here, we reported that *P. rubrigenum* is the dominant symbiotic fungus found in the TL of "Guan Xiang" agarwood formed by the traditional physical wounding method in Guangdong Province, China. This represents a key source of persistent agarwood accumulation with a large change in the metabolism of materials. We hypothesized that *P. rubrigenum* plays a key role in persistent agarwood accumulation. We found that *P. rubrigenum* continuously promoted the production of sesquiterpenes in *A. sinensis* seedlings to a higher degree than the other six fungal species isolated from the agarwood. Interestingly, *C. parahalotolerans*, *C. cladosporioides,* and *Pm. janthinellum*, which is mainly distributed in the NL, rarely induce sesquiterpenes and chromones, which are representative agarwood compounds. Additionally, *L. theobromae*, another fungus mainly distributed in the DL, was able to induce the accumulation of chromones, which is consistent with a previous report (20). Therefore, different fungi have various ways to interact with the *A. sinensis* host tree, which correspond to different abilities to promote agarwood formation. In addition, *P. rubrigenum* has the great ability to promote agarwood sesquiterpene accumulation. Therefore, a mixed fermentation media of *L. theobromae* and *P. rubrigenum* might be a potential way to promote a large amount of sesquiterpene and chromone accumulation in *A. sinensis*, which could improve the agarwood yield.

To address the mechanism of induction of *P. rubrigenum* to its host *A. sinensis*, *A. sinensis* calli stressed by *P. rubrigenum* for different periods were tested. Analysis of sesquiterpene products and biosynthetic proteins revealed that *P. rubrigenum* mainly induced agarwood sesquiterpene biosynthesis in *A. sinensis* through the MVA pathway. To further investigate the mechanism of agarwood sesquiterpene promotion, the protein levels of TFs in *A. sinensis* were explored. However, most TFs showed no significant changes in expression. Therefore, we hypothesized that posttranscriptional modifications might occur in *A. sinensis* TFs. Interestingly, the protein phosphorylation levels of 52.9% of TFs were significant changed, including 20% upregulated phosphoralated TFs and 32.9% downregulated phosphoralated TFs. Correlation analysis between sesquiterpene biosynthetic pathway enzymes and

phosphoralated TFs revealed that phosphoralated MYB, bZIP, and WRKY TFs were highly correlated with the sesquiterpene biosynthetic pathway enzymes.

In summary, our study is the first to report *P. rubrigenum* as one of the dominant fungi in "Guan Xiang" agarwood and to verify its function in promoting persistent agarwood sesquiterpene accumulation. Additionally, the phosphorylation of TFs is the key molecular mechanism of agarwood accumulation through the interaction between *P. rubrigenum* and its host plant *A. sinensis*. Our work is the first to provide molecular data regarding the interaction between *P. rubrigenum* and its host plant *A. sinensis*. This will facilitate further investigations into the molecular mechanisms of fungal infections and plant immunity in *Aquilaria* host trees.

## MATERIALS AND METHODS

**Plant and fungal materials.** Three 50-year-old *A. sinensis* trees in Dalingshan town, Guangdong province, China (latitude 22°45′43″N, longitude 113°48′45″E), were physically wounded using a machete, as described in a previous study (7). After 5 years, 0.5~1.0 cm thick agarwood sections formed beneath the wounded surface. Three trees approximately 30 to 40 cm in diameter were wounded in the transverse section of the main trunks ~1.0 m from the ground.

*A. sinensis* seeds were surface-sterilized with 70% ethanol for 30 s and 2% sodium hypochlorite solution for 15 min and then rinsed five times in sterile water. The seeds were then placed in sterilized soil and germinated at 25℃ in a growth chamber with a 16 h photoperiod. After 1 year of cultivation, the *A. sinensis* seedlings were holed and infected with or without agarwood isolated fungi, including *P. rubrigenum*, *Lasiodiplodia theobromae*, *Trichoderma harzianum*, *Trichoderma atroviride*, *Cladosporium cladosporioides*, *Cladosporium parahalotolerans*, and *Peniciullium janthinellum*. The aforementioned experiments were conducted four times. The infected and uninfected stems of *A. sinensis* seedlings were collected to analyze volatile compounds using GC-MS.

*A. sinensis* calli were induced as described previously (33). The untreated calli were used as the control, whereas the experimental calli were treated with the *P. rubrigenum* LX2018 strain for 3 and 18 days. The three groups were sampled for the analysis. Fresh callus material was harvested, weighed, frozen immediately in liquid nitrogen, and stored at −80℃ until required. All experiments were conducted in triplicate. Untreated and treated *A. sinensis* calli were collected for quantitative proteomics and phosphoproteomic analysis.

**Light and fluorescence microscopy.** Fresh agarwood samples were washed in distilled water, trimming into small pieces, and sections (DL, DA, AL, TL and NL) were cut at a thickness of ~20 $\mu$m using a freezing stage sliding microtome and washed with phosphate-buffered saline (PBS). For light microscopic observation of starch grains and brown resin-like materials, sections were stained with iodine-potassium iodide (I₂-KI) for 3 min. Similarly, sections were stained with periodic acid-Schiffe's reagent (PAS) for 10 to 20 min to investigate the change of polysaccharides. To observe the nuclei, the sections were stained with DAPI (1 $\mu$g/mL in PBS) for in the dark 10 min at room temperature. After staining, the sections were washed with PBS three times, covered with coverslips, and observed under a fluorescence microscopy (Olympus AX 80, Japan).

To analyze the tissue and cell structures of uninfected and *P. rubrigenum*-infected *A. sinensis*, paraffin sections (10-$\mu$m thickness) were obtained using a Thermo Scientific MicRoM HM 325 sliding microtome. For light microscopic observations, sections were stained with Fast Green stain. After staining, the sections were washed with PBS three times, dehydrated with alcohol (50, 80, 90, 95, and 100%), cleared with xylene, sealed with neutral gum, and observed using a fluorescence microscope (Zeiss AX10).

**Extraction of volatile oil.** The dried powders from different layers of agarwood and variously treated stems (0.1 g) were weighed and placed in a 2 mL centrifuge tube. Ethyl acetate (1.5 mL) was then added to the tubes. The sample was incubated overnight at room temperature and extracted for 45 min using the 40 kHz ultrasonic cold extraction method. The upper solvent phase was separated by centrifugation at 12,000 rpm for 10 min at 4℃. After adding ethyl acetate to supplement weightlessness, the volatile oil was filtered using a 0.22 $\mu$m PTFE filter membrane and then stored in a dark glass bottle at 4℃ prior to GC-MS analysis.

**GC-MS analysis.** GC-MS analysis was performed with a Thermo TRACE1310/TSQ 8000 GC-MS (USA) equipped with an HP-5 capillary column (50 m × 0.25 mm internal diameter; 0.25-$\mu$m film thickness) and a mass spectrometer with an ion-trap detector in the full-scan mode under electron impact ionization (70 eV). Helium was used as the carrier gas at a 1 mL · min$^{-1}$ flow rate. Separation was performed under the following conditions: an initial temperature of 50℃ for 1 min, ramped to 150℃ at 10℃·min$^{-1}$, held at 150℃ for 15 min, ramped to 280℃ at 8℃·min$^{-1}$, and held at 280℃ for 10 min. The relative contents of individual components in different sections of agarwood and variously treated stems were expressed as the percent peak area relative to the total peak area. Identification of these constituents was based on comparing retention times and mass spectra with standards or mass spectra in the Wiley or National Institute of Standards and Technology 11 (NIST11) libraries combined with the Kovats retention index (RI). For the determination of RI, a series of n-alkanes (C9-C23) was used under the same experimental conditions. For all GC-MS analyses, six biological replicates were used to analyze the volatile oil in samples.

**Diversity of fungal communities.** Five layers of agarwood samples (DL, DA, AL, TL, NL) were collected, and total genomic DNA was extracted using hexadecyl trimethyl ammonium bromide (CTAB). The ITS1 gene was amplified in all samples using the universal ITS5-1737F primers with a barcode used as a marker to distinguish samples. PCR was performed using the Phusion High-Fidelity PCR Master Mix (New England Biolabs, Ipswich, MA, UK). The PCR products were mixed in equimolar ratios. The mixed PCR products were purified using the Qiagen Gel Extraction kit (Qiagen, Germany). Sequencing libraries were generated using the TruSeq DNA PCR-Free Sample Preparation kit (Illumina, USA), following the manufacturer's recommendations, and

index codes were added. The quality of the libraries was assessed using a Qubit 2.0 Fluorometer (Thermo Scientific) and an Agilent Bioanalyzer 2100 system. High-quality libraries were sequenced on an Illumina HiSeq 2500 platform, which generated 250 bp paired-end reads. High-quality sequences were clustered into operational taxonomic units (OTUs) defined at 97% similarity (34). These OTUs were applied to diversity, richness, and rarefaction curve analyses using MOTHUR software. OTUs that reached 97% similarity were used to make taxonomic assignments using the Quantitative Insights into Microbial Ecology (QIIME) software package (35) and were compared to the SILVA, Greengene, and RDP databases. Venn diagrams were generated to identify the shared and specific taxa between groups using http://www.ehbio.com/test/venn/.

**Fungal isolates and identification.** The agarwood samples were surface-sterilized with 70% ethanol for 30 s and 2% sodium hypochlorite solution for 1 min, rinsed five times in sterile water, and dried on sterile filter paper. Small pieces of agarwood samples were cut just below the surface, plated onto malt extract agar (MEA, Coolaber Science & Technology, Beijing, China), and incubated at 25°C in the dark for 5 to 10 days. Hyphal tips that grew from the agarwood samples were subcultured onto fresh plates. After 3 to 5 rounds of purification, the fungal strains were identified by morphological characteristics and sequence analysis.

**Detecting the volatile oil of *P. rubrigenum* hyphae and fermentation media.** Detailed protocols on the detecting the volatile of *P. rubrigenum* hyphae and fermentation media are available in the Supplementary Methods.

**Quantitative proteomics and phosphoproteome.** Detailed protocols on the quantitative proteomics and phosphoproteome are available in the Supplemental material.

**Data availability.** The sequencing data have been deposited in the Sequence Read of the NCBI under the BioProject accession numbers PRJNA842211.

## SUPPLEMENTAL MATERIAL

Supplemental material is available online only.

**SUPPLEMENTAL FILE 1**, PDF file, 2.5 MB.

## ACKNOWLEDGMENTS

We thank Chongjiu Li from China Agricultural University for the chemical analysis. The authors are also grateful to Ou Huang, the general manager of Guangdong Shangzhengtang Group Co., for his assistance during the agarwood sample collection. This work was supported by the CACMS Innovation Fund (Grant No. CI2021A04101), and the Fundamental Research Funds for the Central Public Welfare Research Institutes (grant no. ZZ13-YQ-095, ZZ13-YQ-093-C1, ZZXT202112).

We have no conflict of interest to declare.

J.L. and L.H. designed the research; J.L., T.L., and X.Z. performed research; J.L., T.C., J.G., and J.Z. analyzed the microbial sequences, quantitative proteomics and phosphoproteome; X.Z. AND T.W. isolated and identified the fungi; J.L., T.L., X.Z., and J.Y. analyzed the chemical data; C.J., X.C., and M.C. collected the agarwood samples; J.L. wrote the paper; J.L., T.C., and L.H. reviewed the paper.

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
