## [Reviewer comments · Microbiology Spectrum]

Microbiology Spectrum

Integrating multiple omics identifies *Phaeoacremonium rubrigenum* acting as *Aquilaria sinensis* marker fungus to promote agarwood sesquiterpene accumulation by inducing plant host phosphorylation

Juan Liu, Tianxiao Li, Tong Chen, Jiaqi Gao, Xiang Zhang, Chao Jiang, Jian Yang, Junhui Zhou, Tielin Wang, Xiulian Chi, Meng Cheng, and Luqi Huang

Corresponding Author(s): Juan Liu, China Academy of Chinese Medical Sciences

Review Timeline:

Submission Date:	December 21, 2021
Editorial Decision:	April 8, 2022
Revision Received:	May 26, 2022
Accepted:	June 6, 2022

Editor: Giuseppe Ianiri

Reviewer(s): Disclosure of reviewer identity is with reference to reviewer comments included in decision letter(s). The following individuals involved in review of your submission have agreed to reveal their identity: Santiago Gutierrez (Reviewer #1)

Transaction Report:

DOI: <https://doi.org/10.1128/spectrum.02722-21>

April 8, 2022

Dr. Juan Liu
China Academy of Chinese Medical Sciences
No. 16, Inner Dongzhimen Southern Small Street
Beijing
China

Re: Spectrum02722-21 (Integrating multiple omics identifies *Phaeoacremonium rubrigenum* acting as *Aquilaria sinensis* marker fungus to promote agarwood sesquiterpene accumulation by inducing plant host phosphorylation)

Dear Dr. Juan Liu:

Thank you for submitting your manuscript to Microbiology Spectrum. Two experts reviewed your manuscript and found several issues. While reviewer 1 raised minor points, reviewer 2 recommended rejection of the manuscript. I believe that the manuscript is overall valid, but I strongly recommend to respond to the 3 main points raised by reviewer 2, in particular if *Pm. rubrigenum* alone is able to produce sesquiterpenes.

Link Not Available

Sincerely,

Giuseppe Ianiri

Journals Department
Reviewer comments:

Reviewer #1 (Comments for the Author):

Article "Integrating multiple omics identifies *Phaeoacremonium rubrigenum* acting as *Aquilaria sinensis* marker fungus to promote agarwood sesquiterpene accumulation by inducing plant host phosphorylation" by Juan Liu and coworkers described a very detailed analysis trying to elucidate the mechanisms of agarwood formation by *Aquilaria sinensis* in interaction with the fungal microbiota. This work was made with *Aquilaria sinensis* host trees of the Guan Xiang area and detected *P. rubrigenum* as an important promoter agent of agarwood sesquiterpene accumulation, which is one of the two major components of agarwood.

This manuscript is well written and clearly presented. It includes a deep analysis of sesquiterpenes detected in the five wood layers, and the relevance of the TL layer for this accumulation. They also found that *Phaeoacremonium rubrigenum* is one of the most abundant species in AL and TL. Authors also describe the secondary metabolites found in the different layer, the predominant fungi found in the different layers, as well as the sesquiterpenes found in them and described some interesting molecular features related with those properties (gene expression, protein phosphorylation,...). Despite these previous considerations, authors can be required to improve clarity in the presentation of some data, and some figures could be shortened to reach this objective

Some minor points to be improved are included below,

Abstract

-Authors use an unusual way to abbreviate *Phaeoacremonium* as Pm., when the canonical way would be just a P.

Results

_ Fig S1A. The legend of this figure must be improved to include all the details regarding what the different panels correspond to. E.g. no clear details about the different layer are included in panel A of this figure.

_ Description of the different panels in Fig. 2 must be improved. This figure includes many data, that would be relevant for conclusions in the article. I found panels D unnecessary and could be removed. Panel A requires a more detailed explanation. Not clear what are the differences between the different graphs in this panel. Thus, it is difficult to reach a conclusion.

_ Fig. 3F. Please indicate in the legend to this figure what layers were analyzed in the lower panels.

_ Section 2.4, Figure 4 panel E. Values included in this panel correspond to protein or transcripts levels?. Authors mixed in this section two very different concepts expression and proteins, proteins are produced no expressed. Genes are expressed. Please clarify and correct.

_ Authors need to revise the English to correct some misspelling words.

Reviewer #2 (Comments for the Author):

In this manuscript, Liu et al. carried out extensive -omics-based studies to understand the interactions between *Pm. rubrigenum* and *A. sinensis*. The study itself is interesting to some extent but needs to be significantly developed before it can be accepted by *Microbiology Spectrum*. There are some places in the manuscript that are confusing to read and need further polishing.

A key question that was not answered in this manuscript, but is generally critical in the plant-microbe interactions, is to determine the compounds of interest are synthesized by which party. The authors have carried out extensive omics studies to show that sesquiterpene production is enhanced in *A. sinensis*, but it will also be helpful to investigate if these terpene-related genes are in *Pm. rubrigenum*. There are two potential scenarios that could be interesting: i) *Pm. rubrigenum* provides precursors to facilitate the production of sesquiterpene in *A. sinensis*, or ii) *Pm. rubrigenum* contains the terpene synthase genes. Therefore, in Line 185-190, The control experiment to conclude that *Pm. rubrigenum* was key to sesquiterpene production was not conclusive. It is likely that *Pm. rubrigenum* itself may produce these sesquiterpenes in the absence of *A. sinensis* seedlings.

Line 146, it is unclear how metabolism in TL changed a lot, is it a comparison with TL at a different time point or with AL and NL? Since no nuclei were observed in AL, the fungi species detected in AL do not represent the ones when AL cells were alive.

In Section 2.4, the proteome analysis of *A. sinensis*: it is unclear how the sampling of *A. sinensis* was carried out. Was *A. sinensis* one-year seedlings? Which layer (AL, NL, or TL) was examined with proteomics?

There are some grammatical errors throughout the manuscript and are not limited to the following:

Line 30, "concerning" should be "for";

Line 36 "sesquiterpene promotion" should be "enhanced sesquiterpene production";

Line 135, "furtherly" should be "further".

Staff Comments:

Preparing Revision Guidelines

Please return the manuscript within 60 days; if you cannot complete the modification within this time period, please contact me. If you do not wish to modify the manuscript and prefer to submit it to another journal, please notify me of your decision immediately so that the manuscript may be formally withdrawn from consideration by Microbiology Spectrum.

Reviewers' comments and point-to-point response:

Reviewer #1 (Comments for the Author):

Article "Integrating multiple omics identifies *Phaeoacremonium rubrigenum* acting as *Aquilaria sinensis* marker fungus to promote agarwood sesquiterpene accumulation by inducing plant host phosphorylation" by Juan Liu and coworkers described a very detailed analysis trying to elucidate the mechanisms of agarwood formation by *Aquilaria sinensis* in interaction with the fungal microbiota. This work was made with *Aquilaria sinensis* host trees of the Guan Xiang area and detected *P. rubrigenum* as an important promoter agent of agarwood sesquiterpene accumulation, which is one of the two major components of agarwood.

This manuscript is well written and clearly presented. It includes a deep analysis of sesquiterpenes detected in the five wood layers, and the relevance of the TL layer for this accumulation. They also found that *Phaeoacremonium rubrigenum* is one of the most abundant species in AL and TL. Authors also describe the secondary metabolites found in the different layer, the predominant fungi found in the different layers, as well as the sesquiterpenes found in them and described some interesting molecular features related with those properties (gene expression, protein phosphorylation, ...). Despite these previous considerations, authors can be required to improve clarity in the presentation of some data, and some figures could be shortened to reach this objective.

Thank you for the reviewer's advice. We have carefully polished our manuscript, and Figure 2 have been shortened to improve the presentation of our data.

Some minor points to be improved are included below,

1. Abstract

Authors use an unusual way to abbreviate *Phaeoacremonium* as *Pm.*, when the canonical way would be just a *P.*

Thank you for the reviewer's advice. We have reviewed *Pm.* as *P.* in the whole manuscript.

2. Results

(1) Fig S1A. The legend of this figure must be improved to include all the details regarding what the different panels correspond to. E.g. no clear details about the different layer are included in panel A of this figure.

Thanks for the detailed advice. We have reviewed the legend of this figure as follows.

Fig. S1. Light and fluorescent microscopic features of fresh agarwood samples.

(A) Micrographs of different sections, showing the distribution pattern of starch grains, polysaccharides, brown resin-like materials and live cells. AL, the agarwood layer; TL, the agarwood-normal transition layer; NL, the normal layer. I section, partial DL and partial DA; II section, partial AL and partial TL; III section, TL; IV section, NL. A-1 shows the brown resin-like materials and starch grains in I section by using I₂-KI staining; A-2 shows the brown resin-like materials and polysaccharides in I section by using Periodic Acid Schiff (PAS) staining; A-3 shows the brown resin-like materials and nuclei in I section by using DAPI staining;

B-1 shows the brown resin-like materials and starch grains in II section by using I₂-KI staining; B-2 shows the brown resin-like materials and polysaccharides in II section by using Periodic Acid Schiff (PAS) staining; B-3 shows the brown resin-like materials and nuclei in II section by using DAPI staining; C-1 shows the brown resin-like materials and starch grains in III section by using I₂-KI staining; C-2 shows the brown resin-like materials and polysaccharides in III section by using PAS staining; C-3 shows the brown resin-like materials and nuclei in III section by using DAPI staining; D-1 shows the brown resin-like materials and starch grains in IV section by using I₂-KI staining; D-2 shows the brown resin-like materials and polysaccharides in IV section by using PAS staining; D-3 shows the brown resin-like materials and nuclei in IV section by using DAPI staining. White arrowheads indicate nuclei; red arrowheads indicate brown resin-like materials; blue arrowheads indicate starch grains; black arrowheads indicate polysaccharides. Scale bars = 100 μm. **(B)** A diagram of distribution patterns of storage starch, polysaccharides, brown resin-like materials and nuclei in different layers of wounded wood. DL, the decay layer; DA, the decay-agarwood transition layer; AL, the agarwood layer; TL, the agarwood-normal transition layer; NL, the normal layer. Label I shows the site of micrographs of A1~A3 in Fig. S1A; Label II shows the site of micrographs of B1~B3 in Fig. S1A; Label III shows the site of micrographs of C1~C3 in Fig. S1A; Label IV shows the site of micrographs of D1~D3 in Fig. S1A.

(2) Description of the different panels in Fig. 2 must be improved. This figure includes many data, that would be relevant for conclusions in the article. I found panels D unnecessary and could be removed. Panel A requires a more detailed explanation. Not clear what are the differences between the different graphs in this panel. Thus, it is difficult to reach a conclusion.

Thanks for the constructive advice. We removed panel D to Fig. S4. We also added the explanation of Fig.2A in the manuscript, and reviewed the legend of this figure as follows.

Manuscript L152-158.

The alpha diversity of fungal species observed was significantly higher in DL compared with the other four layers ($P < 0.05$) (Fig. 2A and Table S3). The community richness indices, including ACE, Chao 1, and PD whole tree, indicated that the three layers containing agarwood resin (DA, AL, TL) possessed the lowest richness. In contrast, DL possessed the richest fungal community (Fig. 2A and Table S3). The Shannon index indicated the lowest fungal diversity in AL and TL, suggesting that the fungal specificities of AL and TL were strong (Fig. 2A and Table S3).

The legend of Figure 2

Fig. 2. Different composition of fungal communities in five agarwood layers.

(A) Boxplot showing alpha diversity estimate, including Observed species, ACE, Chao1, PD whole tree, and Shannon of fungal communities of each layer of agarwood. Cross-sample differences were analyzed using Tukey analysis (Table

S3). Significant differences (p-value < 0.05) across different compartments were indicated with lowercase letters (a, significant difference between DL and the other layers using Tukey analysis, p value < 0.05; b, significant difference between DA and the other layers using Tukey analysis, p value < 0.05; c, significant difference between AL and the other layers using Tukey analysis, p value < 0.05; d, significant difference between TL and the other layers using Tukey analysis, p value < 0.05; e, significant difference between NL and the other layers using Tukey analysis, p value < 0.05). Significant differences (p value < 0.01) across different compartments were indicated with uppercase letters (A, significant difference between DL and the other layers using Tukey analysis, p value < 0.01; B, significant difference between DA and the other layers using Tukey analysis, p value < 0.01; C, significant difference between AL and the other layers using Tukey analysis, p value < 0.01; D, significant difference between TL and the other layers using Tukey analysis, p value < 0.01; E, significant difference between NL and the other layers using Tukey analysis, p value < 0.01). DL, decay layer; DA, decay-agarwood transition layer; AL, agarwood layer; TL, agarwood-normal transition layer; NL, normal layer. **(B)** Unconstrained PCoA (for principle coordinates PCo1 and PCo2) with weighted UniFrac distance depicting the fungal species composition structure differences among different layers of agarwood. **(C)** LEfSe generated the taxonomic cladogram showing marker fungal species of each layer of agarwood. Each dot represents one taxon, with sizes indicating average abundances. The color of dots represents taxon marker belongings of groups. *Phaeoacremonium rubrigenum* was labeled with red color, which was the biomarker of the TL.

(3) Fig. 3F. Please indicate in the legend to this figure what layers were analyzed in the lower panels.

Thanks for the kind advice. We have added the lower panels in Fig. 3F and the corresponding figure legend.

Fig. 3 (F) Comparing relative content of total sesquiterpene and chromone accumulation differences in one-year-old *A. sinensis* seedlings after treatment with six other fungal species for 30 days, including *L. theobromae* (Lth), *T. harzianum* (Tha), *T. atroviride* (Tat), *C. cladosporioides* (Ccl), *C. parahalotolerans* (Cpa) and *Penicillium janthinellum* (Pja). f1: Culture characteristics of the above six fungal species isolated from agarwood on potato dextrose agar (PDA) medium. f2: Pie charts showing the relative abundance of the above fungal genera in each agarwood layer, including *Lasiodiplodia* sp., *Trichoderma* sp., *Cladosporium* sp., *Penicillium* sp. f3: Comparing the total sesquiterpene content in one-year-old *A. sinensis* seedlings after treatment using the holing technique with *P. rubrigenum* (Pmr) and other isolated fungi (Lth, Tha, Tat, Ccl, Cpa, Pja) for 30 days. n = 4. ** p value < 0.01. f4: Comparing the total chromone content in one-year-old *A. sinensis* seedlings after treatment using the holing technique with *P. rubrigenum* (Pmr) and other isolated fungus (Lth, Tha, Tat, Ccl, Cpa, Pja) for 30 days. n = 4. ** p value < 0.01.

(4) Section 2.4, Figure 4 panel E. Values included in this panel correspond to protein or transcripts levels? Authors mixed in this section two very different concepts expression and proteins, proteins are produced no expressed. Genes are expressed. Please clarify and correct.

Thanks to the reviewer for making this clear. Values included in figure 4 E correspond to protein levels. We have corrected misrepresentations in section 2.4 and figure 3E.

Fig. 4 (E) The relative abundance of enzymes involved in sesquiterpene biosynthesis by using quantitative proteomics analysis, including mevalonate (MVA) pathway and methylerythritol 4-phosphate (MEP) pathway. AACT, acetyl-CoA C acetyltransferase; CME, 4-(cytidine 5'-diphospho)-2-C-methyl-D-erythritol; CMEC, 2-C-methyl-D-erythritol 2,4-cyclodiphosphate; CMK, 4-(cytidine 5'-diphospho)-2-C-methyl-D-erythritol kinase; DXP, 1-deoxy-D-xylulose 5-phosphate; DXR, 1-deoxy-D-xylulose 5-phosphate reductoisomerase; DXS, 1-deoxy-D-xylulose 5-phosphate synthase; FPS, farnesyl diphosphate synthase; G3P, glyceraldehyde 3-phosphate; HDS, 4-hydroxy-3-methylbut-2-enyl diphosphate synthase; HDR, 4-hydroxy-3-methylbut-2-enyl diphosphatereductase; HMED, 4-hydroxy-3-methylbut 2-enyl diphosphate; HMG-CoA, hydroxyl methyl glutaryl-CoA; HMGS, hydroxyl methylglutaryl-CoA synthase; HMGR, hydroxyl methyl glutaryl-CoA reductase; IPP, isopentenyl diphosphate; IPPI, isopentenyl diphosphate isomerase; MCS, 2-C-methyl-D-erythritol 2,4-cyclodiphosphate synthase; MCT, 2-C-methyl-D-erythritol-4-phospate cytidyltransferase; MEP, 2-C-methyl-D-erythritol 4-phosphate; MVA, mevalonate; MVA-5P, mevalonate-5-phosphate; MVA-5PP, mevalonate-5-diphosphate; MVD, mevalonate pyrophosphate decarboxylase; MVK, mevalonate kinase; PCME, 2-phospho-4-(cytidine 5'-diphospho)-2-C-methyl-D-erythritol; PMK, 5-phosphomevalonate kinase; SesTPS1 and SesTPS2, guaiane synthases; SesTPS3, humulene synthase; SesTPS4, nerolidol synthase. n = 3.

(5) Authors need to revise the English to correct some misspelling words.

Thanks for the reviewer's advice. We have reviewed the whole manuscript for wrong spellings, and we also searched for help from Editage for further polishing the manuscript.

Editing Certificate

This document certifies that the manuscript listed below has been edited to ensure language and grammar accuracy and is error free in these aspects. The edit was performed by professional editors at Editage, a division of Cactus Communications. The author's core research ideas were not altered in any way during the editing process. The quality of the edit has been guaranteed, with the assumption that our suggested changes have been accepted and the text has not been further altered without the knowledge of our editors.

MANUSCRIPT TITLE

Integrating multiple omics identifies *Phaeoacremonium rubrigenum* acting as an *Aquilaria sinensis* marker fungus to promote agarwood sesquiterpene accumulation by inducing plant host phosphorylation

AUTHORS

Juan Liu, Tianxiao Li, Tong Chen, Jiaqi Gao, Xiang Zhang, Chao Jiang, Jian Yang, Junhui Zhou, Tielin Wang, Xiulian Chi, Meng Cheng, Luqi Huang

ISSUED ON

May 20, 2022

JOB CODE

ZYJUA_12

Vikas Narang

Vikas Narang
Chief Operating Officer - Editage

Reviewer #2 (Comments for the Author):

In this manuscript, Liu et al. carried out extensive -omics-based studies to understand the interactions between *Pm. rubrigenum* and *A. sinensis*. The study itself is interesting to some extent but needs to be significantly developed before it can be accepted by Microbiology Spectrum. There are some places in the manuscript that are confusing to read and need further polishing.

(1) A key question that was not answered in this manuscript, but is generally critical in the plant-microbe interactions, is to determine the compounds of interest are synthesized by which party. The authors have carried out extensive omics studies to show that sesquiterpene production is enhanced in *A. sinensis*, but it will also be helpful to investigate if these terpene-related genes are in *Pm. rubrigenum*. There are two potential scenarios that could be interesting: i) *Pm. rubrigenum* provides precursors to facilitate the production of sesquiterpene in *A. sinensis*, or ii) *Pm. rubrigenum* contains the terpene synthase genes. Therefore, in Line 185-190, the control experiment to conclude that *Pm. rubrigenum* was key to sesquiterpene production was not conclusive. It is likely that *Pm. rubrigenum* itself may produce these sesquiterpenes in the absence of *A. sinensis* seedlings.

Thanks for the reviewer's insightful advice. Just as the reviewer said, the sesquiterpenes might be produced by *P. rubrigenum*. Actually, we first thought *P. rubrigenum* itself might produce these sesquiterpenes in the absence of *A. sinensis*. Thus, we sequenced the genome of *P. rubrigenum* in the year 2018 (unpublished data). However, we didn't find any terpene synthase genes in the genome of *P. rubrigenum*. Thus, we thought *P. rubrigenum* could not produce these sesquiterpenes, but induce *A. sinensis* seedlings to produce sesquiterpenes. To examine this thought, the volatile compounds of *P. rubrigenum* thalli and fermentation both were tested by GC-MS in this study (Fig. S6). No any sesquiterpenes could be detected. FPP, which is the precursors of sesquiterpenes, could also not be detected. Thus, it should be the *A. sinensis* seedling to produce agarwood sesquiterpene instead of *P. rubrigenum*. Thus, we mainly focused on addressing the mechanism of the induction of *P. rubrigenum* to its host *A. sinensis*.

Fig. S6. Gas chromatography-mass spectroscopy profiles of *P. rubrigenum* hyphae and

fermentation liquor. (A) Total ion chromatogram of the volatile compounds of 5-day cultured *P. rubrigenum* hyphae and fermentation liquor. n=3. There is no any sesquiterpenes and FPP detected. **(B)** Total ion chromatogram of the volatile compounds of 10-day cultured *P. rubrigenum* hyphae and fermentation liquor. n=3. There is no any sesquiterpenes and FPP detected.

(2) In Section 2.4, the proteome analysis of *A. sinensis*: it is unclear how the sampling of *A. sinensis* was carried out. Was *A. sinensis* one-year seedlings? Which layer (AL, NL, or TL) was examined with proteomics?

Thanks for the advice. The proteome analysis of *A. sinensis* was carried out by using the *A. sinensis* calli. The reason why we choose the *A. sinensis* calli to detect the proteome analysis is that the callus system of *A. sinensis* is more stable than the other systems. We have further revised this section.

(3) There are some grammatical errors throughout the manuscript and are not limited to the following:

---Line 30, "concerning" should be "for";

Revised.

---Line 36 "sesquiterpene promotion" should be "enhanced sesquiterpene production";

Revised.

---Line 135, "furtherly" should be "further".

Revised.

We have revised the grammar of our manuscript thoroughly by ourselves and also by Editage for polishing the manuscript.

June 6, 2022

Dr. Juan Liu
China Academy of Chinese Medical Sciences
No. 16, Inner Dongzhimen Southern Small Street
Beijing
China

Re: Spectrum02722-21R1 (Integrating multiple omics identifies *Phaeoacremonium rubrigenum* acting as *Aquilaria sinensis* marker fungus to promote agarwood sesquiterpene accumulation by inducing plant host phosphorylation)

Dear Dr. Juan Liu:

Your manuscript has been accepted, and I am forwarding it to the ASM Journals Department for publication. You will be notified when your proofs are ready to be viewed.

Sincerely,

Giuseppe Ianiri
Editor, Microbiology Spectrum
